# Ablation of the Integrin CD11b Mac-1 Limits Deleterious Responses to Traumatic Spinal Cord Injury and Improves Functional Recovery in Mice

**DOI:** 10.3390/cells13181584

**Published:** 2024-09-20

**Authors:** Yun Li, Zhuofan Lei, Rodney M. Ritzel, Junyun He, Simon Liu, Li Zhang, Junfang Wu

**Affiliations:** 1Department of Anesthesiology and Center for Shock, Trauma and Anesthesiology Research (STAR), University of Maryland School of Medicine, 685 W. Baltimore Street, MSTF, Room 6-034D, Baltimore, MD 21201, USA; yun.li@som.umaryland.edu (Y.L.); zlei@som.umaryland.edu (Z.L.); rodney.m.ritzel@uth.tmc.edu (R.M.R.); junyunhe0123@gmail.com (J.H.); simonliu@mednet.ucla.edu (S.L.); 2Department of Neurology, McGovern Medical School, The University of Texas Health Science Center at Houston, Houston, TX 77030, USA; 3Department of Physiology, Center for Vascular and Inflammatory Diseases, University of Maryland School of Medicine, Baltimore, MD 21201, USA; lizhang@som.umaryland.edu

**Keywords:** spinal cord injury, Mac-1/CD11b, microglia/monocytes, neuroinflammation

## Abstract

Spinal cord injury (SCI) triggers microglial/monocytes activation with distinct pro-inflammatory or inflammation-resolving phenotypes, which potentiate tissue damage or facilitate functional repair, respectively. The major integrin Mac-1 (CD11b/CD18), a heterodimer consisting of CD11b and CD18 chains, is expressed in multiple immune cells of the myeloid lineage. Here, we examined the effects of CD11b gene ablation in neuroinflammation and functional outcomes after SCI. qPCR analysis of C57BL/6 female mice showed upregulation of *CD11b* mRNA starting from 1 d after injury, which persisted up to 28 d. CD11b knockout (KO) mice and their wildtype littermates were subjected to moderate SCI. At 1 d post-injury, qPCR showed increased expression of genes involved with inflammation-resolving processes in CD11b KO mice. Flow cytometry analysis of CD45^int^Ly6C^−^CX3CR1^+^ microglia, CD45^hi^Ly6C^+^Ly6G^−^ monocytes, and CD45^hi^Ly6C^+^Ly6G^+^ neutrophils revealed significantly reduced cell counts as well as reactive oxygen species (ROS) production in CD11b KO mice at d3 post-injury. Further examination with NanoString and RNA-seq showed upregulation of pro-inflammatory genes, but downregulation of the ROS pathway. Importantly, CD11b KO mice exhibited significantly improved locomotor function, reduced cutaneous mechanical/thermal hypersensitivity, and limited tissue damage at 8 weeks post-injury. Collectively, our data suggest an important role for CD11b in regulating tissue inflammation and functional outcome following SCI.

## 1. Introduction

Spinal cord injury (SCI) is a devastating event that leads to long-term disability and even life-threatening consequences. After the initial mechanical damage of the spinal cord, secondary injury processes produce a cascade of biochemical changes within the lesion site and surrounding tissue. These delayed and often long-lasting processes include activation of microglia and astrocytes, infiltration of macrophages and other peripheral immune cells, overproduction of reactive oxygen species (ROS), and neurotoxicity [1,2,3], leading to extended tissue loss and neurological dysfunction. Following activation, microglia/macrophages present distinct disease-associated phenotypes shortly after injury: pro-inflammatory and inflammation-resolving [4]. The former can exacerbate tissue damage, while the latter can potentiate functional recovery. This process is driven by the complex interaction between the central nervous system (CNS)’s resident microglia and infiltrated immune cells. However, the underlying mechanisms for phenotype determination are still not fully understood.

Many studies have demonstrated that integrins, a large family of transmembrane proteins, regulate the interaction between immune cells and their extracellular environment, thus playing a major role in driving the polarization of macrophages and microglia [5,6,7,8]. Examples include microglia/macrophage beta2 integrins binding to vascular adhesion molecules in the vascular lumen and promoting migration toward the injury site [9]. Using monoclonal antibodies, studies showed that blocking integrin αDβ2 (CD11d/CD18) activity reduced intraspinal inflammation, improved neurological functions, and was able to attenuate systemic inflammatory syndrome after SCI [9,10,11,12]. Moreover, β1-integrin has been reported to regulate astrogliosis and glial scar formation following SCI [13], suggesting that immune modulation through the integrin family is a potential route for future strategies. More recently, studies have shown that inhibition of integrin β1 and its downstream signaling pathway can improve blood spinal cord barrier repair and promote neurological recovery after trauma [14,15].

The major integrin CD11b/CD18 (also known as Mac-1, αMβ2, and CR3) is a heterodimer of CD11b (αM) and CD18 (β2) subunits [16,17]. The CD11b/CD18 integrin is highly expressed in all myeloid lineage immune cells, including microglia, macrophages, and neutrophils. As early as 1999, a study reported on the expression levels of intercellular adhesion molecule 1 (ICAM-1) and CD11b in the acute stages of CNS after mechanical compression, showing significant upregulation in both the lesion center and perifocal zones within the first week of injury [18]. More recently, studies using mouse models of optic nerve injury, hypertensive cardiac remodeling, liver fibrosis, pneumococcal pneumonia, and others have shown that CD11b knockout (KO) can promote debris clearance, attenuate neural degradation, and reduce macrophage infiltration [19,20,21,22,23]. The extensive reporting on the role that CD11b/CD18 integrin plays in regulating myeloid cells has led us to explore the possibility of using CD11b as a therapeutic target in SCI. Prior work has shown that CD11b ablation can have neuroprotective effects in CNS diseases [24,25,26,27]. However, CD11b also mediated anti-inflammatory activity following activated protein C (APC) treatment of sepsis [28,29,30,31]. Thus far, only a few studies have examined the role of Mac-1 in SCI, with one recent paper reporting that Mac-1/CR3 is an important component in the development of injury-induced neuropathic pain [32], while another earlier study found that CD11b contributed to myelin-induced inflammatory response after injury [33]. Overall, the specific role of CD11b in SCI pathophysiology is still understudied and has yet to be fully addressed.

Here, we utilize a contusion mouse SCI model to examine cellular and molecular changes as well as behavior and histopathology in the absence and presence of the CD11b gene. We demonstrate that SCI-induced acute inflammatory response is increased, whereas production of free oxygen species in microglia/macrophages/neutrophils is reduced in the CD11b KO mice, as measured by qPCR, flow cytometry, NanoString, and bulk RNA-seq analysis. Importantly, we report that functional deficits in locomotion and hypersensitivity to mechanical and thermal stimuli are significantly limited in CD11b KO mice and were associated with reduced tissue damage. Together, our studies demonstrate that CD11b regulates the status of beneficial neuroinflammation in the early-phase response to SCI, potentially contributing to SCI pathophysiology.

## 2. Materials and Methods

### 2.1. Animals and Mouse Spinal Cord Contusion Model

All surgical and experimental procedures in this study were performed under protocols approved by the Institutional Animal Care and Use Committee (IACUC) at the University of Maryland School of Medicine. To reduce the risk of bladder infection [34,35], we only used female mice in this study. Young adult (10–12 weeks) female CD11b KO mice were obtained from Dr. Li Zhang’s lab and maintained in the UMB animal facility. Following the induction of anesthesia by isoflurane, a laminectomy was carried out at approximately the T9 to T11 levels, and the spinal column was secured with metal clamps to the lateral processes. The mice were subjected to a midline contusion injury of the spinal cord at the T10 level using the Infinite Horizon Spinal Cord Impactor (Precision Systems and Instrumentation) with a force of 60 kilodyne, classified as moderate injury [36]. Manual bladder expression was carried out thrice daily for the first 7–14 days after SCI until reflex bladder emptying was re-established. For mice used as control animals, sham surgery was performed after anesthesia, consisting of only a laminectomy without contusion. Individuals who performed functional assessment and data analysis were blinded to group designations and genotypes throughout all experiment stages.

### 2.2. Flow Cytometry

Following euthanization, mice were perfused with 40 mL of ice-cold phosphate buffered saline (PBS). A segment (5 mm length) of spinal cord tissue surrounding the lesion area was isolated, weighed, and placed in complete Roswell Park Memorial Institute (RPMI) 1640 (Cat# 22400105, Invitrogen, Waltham, MA, USA) medium with 10% fetal bovine serum (FBS). The tissue samples were mechanically and enzymatically digested in collagenase/dispase (Cat# 10269638001, 1 mg/mL; Roche Diagnostics, Indianapolis, IN, USA), papain (Cat# LS003119, 5 U/mL; Worthington Biochemical, Lakewood, NJ, USA), 0.5 M EDTA (Cat# 15575020, 1:1000; Invitrogen), and DNAse I (Cat# 10104159001, 10 mg/mL; Roche Diagnostics) for 1 h at 37 °C on a shaking incubator (200 rpm). The cell suspension was washed with RPMI and filtered through a 70-µm cell strainer before adding RPMI to a final volume of 3 mL and kept on ice. The samples were then transferred to FACS tubes and washed with FACS buffer, followed by a 10 min incubation period with Fc Block (Cat# 101320, Clone: 93; Biolegend, San Diego, CA, USA) on ice. The spinal cord cells were stained with the following surface antigens: [CD45-eF450 (Cat# 48-0451-82, Clone: 30-F11; eBioscience, Waltham, MA, USA), CD11b-APC/Fire™750 (Cat# 101262, Clone: M1/70; Biolegend), Ly6C-APC (Cat# 128016, Clone: HK1.4; Biolegend), Ly6G-AF700 (Cat# 128024, Clone: 1A8; Biolegend), Cx3CR1-PE (Cat# 149006, Clone: SA011F11, Biolegend)] and Zombie Aqua fixable viability dye (Cat# 423102, Biolegend). The spinal cord cells were washed in FACS buffer, fixed in 2% paraformaldehyde (PFA) for 8 min, and washed once more before adding 500 µL FACS buffer. We used H2DCFDA (DCF, Cat# C6827, 5 µm; Thermo Fisher Scientific, Waltham, MA, USA) to measure ROS production according to the manufacturer’s instructions. As described in previous publications [37,38], DCF was added to RPMI media, vortexed, and incubated for 30 min in a 37 °C water bath. All data were acquired on a BD LSRFortessa cytometer FACSDiva 6.0 (BD Biosciences, Franklin Lakes, NJ, USA) and analyzed using FlowJo v10.10 (Treestar Inc., San Carlos, CA, USA), with a minimum of 5 million events collected for each sample. Countbright^TM^ Absolute Counting Beads (Cat# C36950; Invitrogen, Waltham, MA, USA) were used to estimate cell counts, expressed as total counts/mg spinal cord tissue weight. Leukocytes were first gated using a splenocyte reference (SSC-A vs. FSC-A), followed by singlets (FSC-H vs. FSC-W) and live cell gating based upon exclusion of Zombie Aqua (SSC-A vs. Zombie Aqua-Bv501). Resident microglia were identified as the CD45^int^Ly6C^−^CX3CR1^+^ population, whereas peripheral myeloid cells were identified as the CD45^hi^Ly6C^+^CX3CR1^+^ population. Within the myeloid subset, further separation was achieved by identifying monocytes as Ly6C^hi^Ly6G^−^ and neutrophils as Ly6C^+^Ly6G^+^. Cell type-matched fluorescence minus one (FMO) control was used to determine the positivity of each antibody, and the DCF dye [39].

### 2.3. Total RNA Extraction and Real-Time PCR (qPCR)

RNA samples were obtained after processing with the RNeasy mini kit (Cat# 74104, Qiagen, Tegelen, The Netherlands) from a 5-millimeter-length spinal cord tissue at various post-injury time points. Complementary DNA (cDNA) was synthesized by a Verso cDNA RT kit (Cat# AB1453B, Thermo Scientific, Waltham, MA, USA) per the manufacturer’s protocol. Real-time PCR for target mRNAs was performed with TaqMan gene expression assays for Itgam (Mm00434455_m1), CD83 (Mm01350412_m1), P2yr12 (Mm01950543_s1), Tmem119 (Mm00525305_m1), Trem2 (Mm04209424_g1), Gfap (Mm01253033_m1), Gdf15 (Mm00442228_m1), Opalin (Mm00463365_m1), Fcrls (Mm01219428_m1), Cxcl10 (Mm00445235_m1), Fkbp5 (Mm00487406_m1), H2-T23 (Mm00439246_g1), Csf1r (Mm01266652_m1), Pllp (Mm00452740_m1), Ennp6 (Mm00624107_m1), and Plekhm1 (Mm00805590_m1).

### 2.4. Sample Processing and Western Blot Analysis

At 3 d post-injury, mice spinal cord tissue at a length of 5 mm was extracted from the epicenter of injured WT or CD11b KO mice. For sham animals, an equal length of 5 mm was extracted at the same time from the approximate area of T10. For the processing of tissue samples, RIPA lysis buffer (Cat# R0278, Sigma-Aldrich, Burlington, MA, USA) supplemented with 1 × protease inhibitor cocktail (Cat# P8340, Sigma-Aldrich), Phosphatase Inhibitor Cocktail II (Cat# P5726, Sigma-Aldrich), and Phosphatase Inhibitor Cocktail III (Cat# P0044, Sigma-Aldrich, USA) was used for homogenization on ice. Following sonication and centrifugation at 20,000× *g* for 20 min, protein concentration was determined with the Pierce BCA method (Cat# 23227, Thermo-Fisher Scientific). For SDS-PAGE, samples were run on a criterion TGX 4–20% gradient gel (Bio-Rad, Hercules, CA, USA) and transferred to a 0.2 μm nitrocellulose membrane (Bio-Rad). After blocking with 5% non-fat skim milk in PBST, membranes were incubated overnight in primary antibodies diluted with blocking buffer. The next day, membranes were incubated in HRP-conjugated secondary antibodies and visualized with Supersignal West Pico PLUS Chemiluminescent Substrate (Cat# 34577, Thermo-Fisher Scientific). Images were acquired with the Chemi-Doc MP Imaging system (Bio-Rad) and optical density was quantified with the Image Lab software 6.1 (Bio-Rad). Primary antibodies and their respective dilutions are as follows: NLRP3 (1:500; Cat# 15101S, Cell Signaling Technologies, Danvers, MA, USA), Iba1 (1:1000, Cat# 17198S, Cell Signaling Technologies, Danvers, MA, USA), and β-actin (1:10,000; Cat# SAB1305567A1978, Sigma-Aldrich, Burlington, MA, USA).

### 2.5. NanoString Analysis

RNA samples were acquired from 5 mm of spinal cord tissue one day after SCI. We used the NanoString nCounter^®^ system to analyze total RNA (20 ng/L) and obtain the transcript counts for 757 genes and 13 housekeeping genes from the Mouse Neuroinflammation panel (NanoString Technologies, Seattle, WA, USA). The gene transcript counts were normalized before further analysis and paired differential expression analysis was carried out using NanoString’s nSolver software Version 4.0. The statistical analysis of NanoString data was conducted with R version 4.2.2 in RStudio version 2023.6.0, build 421. Partial least square discriminant analysis (PLS-DA) was performed with the Mixomics package [40], while the online Enrichr toolkit from Maayan Lab was used for pathway enrichment analysis [41,42]. All plots were drawn with the ggplot2 package v3.5.1 [43], with volcano plots and heatmaps being rendered by the EnhancedVolcano v1.22.0 and ComplexHeatmap v2.20.0 [44] packages, respectively.

### 2.6. Bulk RNA Sequencing and Transcriptomic Analysis

Total RNA was extracted using RNeasy mini kit (Qiagen) and sent to Novogen (Sacramento, CA, USA) for RNA quality test and RNA-seq. The RNA quality was initially analyzed using Nanodrop and Agarose Gel Electrophoresis, and the quality of cDNA library is analyzed using an Agilent 2100 Bioanalyzer. RNA-seq transcriptomic analysis of all libraries was performed on Illumina, with data being converted into FastQ format for downstream analysis. The RNA-seq results were validated for selected genes using reverse-transcription real-time quantitative PCR (RT-qPCR). Bioinformatics analyses were performed by Novogen in RStudio version 1.2 with R v3.6.1. The FastQC v0.11.8 package [45] was used for data quality assessment and ribosomal RNA were filtered with SortMeRNA [46]. For quantification of transcript-level abundance, the quasi-mapping-based mode was used in Salmon v1.0.0 [47], with mapping to GRCm38 (mm10) mouse reference genome. Transcript-level abundance was aggregated with gen-level abundance with tximport [48], while the package DESeq2 v1.44.0 [49] was used for differential expression analysis. Volcano plots and heatmaps were rendered with the EnhancedVolcano and ComplexHeatmap [44] packages in Rstudio. Pathway enrichment analysis was performed with the online Enrichr toolkit from Maayan Lab [41,42].

### 2.7. Tissue Processing and Histologic, Immunohistochemistry Analysis

The mouse spinal cord was dissected following intracardial perfusion with ice-cold normal saline and 4% paraformaldehyde. The segments containing the lesion area, or an equal length of spinal cord for sham mice, were embedded in an optimal cutting temperature (O.C.T) compound and cut into 20-µm-thick serial sections. In the assessment of lesion volume, sections spaced 1 mm apart within a segment of 5 mm on each side from rostral to caudal of the injury epicenter were stained with GFAP (1:1000; Cat# Z0334, Dako, Carpinteria, CA, USA) and DAB (Cat# PK-6100, Vector Labs, Newark, CA, USA) as the chromogen. The Cavalieri method was used with a grid spacing of 200 µm in the Stereo Investigator Software version 2022.2.1 (MBF Biosciences, Williston, VT, USA) for quantification as described in previous publications [50]. Areas absent of GFAP are considered part of the lesion cavity, from which the total volume was calculated. For visualizing and quantification of spared white matter (SWM), Luxol fast blue (LFB, Cat# S3382, Sigma-Aldrich) staining was performed as described in previous publications [50,51]. The section with the least amount of SWM was defined as the lesion epicenter, which was imaged at 4× magnification. Using ImageJ 1.54f (RRID: SCR_003070), each section’s total LFB positive area was calculated for SWM assessment and presented as a percentage of the total area.

For immunohistochemistry (IHC) analysis, five coronal spinal cord sections around the lesion site with a 200 μm interval were applied for IHC staining as described in previous publications [52]. Primary antibodies of rabbit Iba1 (1:200; Cat# 17198S, Cell Signaling), rabbit GFAP (1:200; Cat# PA5-16291, Invitrogen), rat CD86 (1:50; Cat# ab955, Abcam, Cambridge, MA, USA), and rat PDGFrB (1:100; Cat# 14-1402-82, Thermo Fisher) were used along with secondary antibodies of Alexa Fluor 488 goat anti-rabbit IgG (1:800, Cat# A11008, Invitrogen) and Alexa Fluor 546 goat anti-rat IgG (1:800, Cat# A21247, Invitrogen). Fluorescent images were obtained through tile scanning with a Nikon Ti-E inverted microscope at 20× magnification (CFI Plan APO VC 20X NA 0.75 WD 1 mm). The images were then analyzed using the NIH ImageJ software (1.54), wherein a counting window of 165.46 um × 165.46 um was randomly selected in the specified region for signal sampling. Two sections per mouse at 400–600 um distance rostrally from the epicenter were used for manual calculation of Iba1/CD86 positive cells. For GFAP/PDGFrB distribution, two sections per mouse close to the epicenter were used, in which the counting windows were applied in the injury core for PDGFrB or in the peri-injury area for GFAP. Identical imaging parameters and data analysis were adopted for all the samples per outcome.

### 2.8. Neurological Behavioral Tests

Basso mouse scale (BMS) for locomotion: For evaluation of hindlimb locomotor function with BMS [53], mice were placed within a level, enclosed arena 100 cm in diameter. Two trained researchers, unaware of the genotype of each mouse, observed them for at least 4 min. The animals were evaluated on a scale of 0 to 9, where 0 indicated total paralysis and 9 was normal activity. The grading criteria are derived from evaluating joint mobility, weight distribution, plantar stepping, and coordination. The mice underwent BMS score testing on day 1, day 3, and at weekly intervals for up to 6 weeks after SCI.

Catwalk XT automated gait analysis: At 7 weeks after SCI, CatwalkXT automated system (Noldus; RRID: SCR_004074) was used for gait analysis as described previously [54,55]. Each mouse underwent a single testing session to maintain situational novelty and encourage exploration of the CatWalk. The CatWalk apparatus features a glass walkway with a light source emitting green LED light, which would be refracted upon contact with the paws. The areas of contact are then captured with a high-speed color camera and recorded in the CatWalk XT v10.6. software. Data acquisition was performed in a darkened room by a single researcher blinded to the mice genotype and handling each subject. Animals were placed in the open end of the CatWalk and allowed to walk across the walkway to the darkened escape enclosure. A minimum of three valid runs, or complete walkway crossings, were obtained for each subject. Trials where the animal stopped partway across or turned around during a run were excluded from the analysis.

Hot plate test: To test SCI-induced cutaneous hypersensitivity of the hind paws, mice were placed on the contact probe of a computerized thermal stimulator on an Incremental Hot/Cold Plate Analgesia Meter (PE34, IITC Life Science, Woodland Hills, CA, USA). The temperature was increased from 30 to 50 °C with the incremental rate at 10 °C per minute. The test was stopped when the tested mouse licked either of its hind paws, and the threshold temperature was recorded. The test was conducted twice with the interval of 3 h, and the average stop temperature and latency time was recorded.

Von Frey test: The von Frey filament method was used to detect hind-paw withdrawal from a mechanical stimulus according to the “up-down” Von Frey method outlined by Caplan et al. [56] and simplified by Bonin et al. [57]. At the start of the experiment, each mouse was individually placed in Plexiglass cubicles on a wire mesh platform and allowed to acclimate for 30 min. The von Frey filaments (MyNeuroLab, St. Louis, MO, USA), with incremental stiffness ranging from 0.04 g to 2.0 g, were applied to the plantar surface of each hind paw. The experiment tests the response of the hind paw, with the first stimulus being set at a force estimated to be close to the 50% withdrawal threshold (0.4 g). A positive response is a brisk paw withdrawal (at least 3 times out of 5 applications) in response to the filament, while no paw withdrawal was considered a negative response. If there is no response, the next filament with a higher force is tested; if there is a response, the next lower-force filament is tested. This continues until 20 readings are obtained for each paw, and the sequence of outcomes (−for no response or +for response) is recorded. The statistical formula published by Dixon et al. was used to determine the mechanical force required to elicit a paw withdrawal response in 50% of animals [58]. The average 50% pain threshold across three trials (one trial per day) was used to estimate the mechanical force threshold accurately.

### 2.9. Statistical Analysis

Quantitative data are all plotted as mean ± standard error of mean, and individual data point is shown. The animal numbers required for each experiment were derived from variability estimated from published data and power calculations based on effect sizes defined by Cohen [59]. All statistical analyses were conducted with GraphPad Prism (Version 9.00) for Windows (RRID: SCR_002798, GraphPad Software, Boston, MA, USA). Two-way ANOVA with repeated measures was used to analyze BMS scores, followed by Sidak’s multiple comparisons post hoc test. One-way or two-way ANOVA was used for multiple comparisons between groups, followed by Tukey’s multiple comparisons post hoc test parametric data (normality and equal variance passed). For lesion volume, stereological data were analyzed using Student’s *t*-test. Statistical analysis in each assay was detailed in figure legends. A *p*-value of ≤0.05 was considered statistically significant. Using a compilation of the Catwalk data, PLS-DA was used to calculate the overall percentage of variation that mice genotype would contribute to gait. For pairwise Spearman correlation analysis, all Catwalk parameters were included with the added rank variable of injury (0 = sham, 1 = SCI), genotype (0 = WT, 1 = CD11b KO), and the final BMS scores at 6w post-injury.

## 3. Results

### 3.1. Ablation of CD11b Gene Limits Acute Proinflammatory Response to SCI

As a first step in exploring the role that CD11b plays in acute neuroinflammation, we used qPCR analysis to examine the changes to its expression level at multiple timepoints after SCI. We observed a significant increase in CD11b mRNA expression starting at 1 d post-injury, reaching its peak at 7 d and remaining persistently high for up to 28 d (Figure 1A). Next, we examined the effects of CD11b ablation on microglia and monocyte markers, along with genes that reflect their phenotypic functions. At 1d after SCI, qPCR analysis showed near-zero levels of CD11b (*Itgam*) mRNA expression in both sham and SCI of CD11b KO mice, as expected (Figure 1B). Furthermore, the gene *Cxcl10*, a chemokine well-known for inducing microglia migration and initiating microglial activation [60], showed significant increases in the injured spinal cord of both genotype groups, with CD11b KO mice showing even higher levels (Figure 1C). In contrast, the gene *Trem2*, which is expressed in microglia and other myeloid cells of the CNS, showed a marked increase in SCI/WT mice, but no injury-induced changes were observed in CD11b KO mice (Figure 1D). As a member of the transforming growth factor beta superfamily and known as macrophage inhibitory cytokine-1 (MIC-1), the expression levels of *Gdf15* were very low in both sham groups but showed significant increases after injury. Pairwise comparison of SCI/WT with SCI/CD11b KO mice showed significantly lower expression levels in the latter group (Figure 1E). Other microglia and myeloid cell markers that we tested included *P2ry12*, *Tmem119*, *CD83*, and *Csf1r*. The markers *P2ry12* (Figure 1F) and *Tmem119* (Figure 1G), both of which are abundantly expressed in ramified microglia [61,62], showed a significant decrease after 1d SCI without genotype effects. The gene *CD83* plays a critical role in controlling and resolving immune responses, showed significantly lower mRNA levels in injury groups (Figure 1H), but no genotypic differences were observed. Finally, the gene that encodes microglial receptor *Csf1r* showed neither injury nor genotype effects (Figure 1I). In addition, both WT and CD11b KO mice showed marked upregulation of *Gfap* at 1 d post-injury, but no genotype effects were observed, suggesting that genetic depletion of CD11b did not affect astrocyte function (Figure 1J). Collectively, acute SCI leads to increased levels of genes that initiate inflammation-resolving processes and drive myeloid cells towards clearance of damaged tissue and debris, which is partially affected by CD11b ablation.

### 3.2. Depletion of CD11b Reduces the Number of Microglia and Infiltration of Neutrophils in the Injured Spinal Cord

At d3 after SCI, when macrophage infiltration peaks in the spinal cord, flow cytometry was used to examine the cellular inflammatory response in the lesion site. Due to the absence of CD11b protein in the CD11b KO mice, we used other known microglia and monocyte markers for gating myeloid cell populations. The cell surface marker CX3CR1 is expressed in microglia and macrophages [63]. As indicated in Figure 2A, microglia were gated on the criteria of CD45 intermediate (int), Ly6C low, and CX3CR1 positive (+), which was validated in WT groups with the traditional gating strategy of CD45^int^CD11b^+^. Although there is a question of whether CD11b would affect microglia survival in sham mice, we observed no differences in the microglia cell counts of WT and CD11b KO mice. For infiltrating myeloid cells, the gating strategy of CD45^hi^Ly6C^+^Ly6G^−^ was used to identify monocytes, while Ly6G^+^ indicated neutrophils. Using this new strategy, we were able to observe a significant increase in microglia cell counts at the lesion site of both WT and CD11b KO mice compared to sham groups of the same genotype; however, the number of microglia were markedly lower in SCI/CD11b KO mice (Figure 2B), suggesting lower levels of injury-induced proliferation. Although the number of infiltrating monocytes remained the same between the two genotypes, CD11b KO mice had significantly lower number of neutrophils (Figure 2C). Furthermore, ROS production was significantly attenuated in both microglia (Figure 2D,E) and infiltrating myeloid cells of SCI/CD11b KO (Figure 2F,G) mice compared to the injured control group, as determined by DCF mean fluorescence intensity. To confirm the anti-inflammatory effects of CD11b KO, we used a separate batch of mice for Western blot analysis at 3d post-injury. After SCI, Western blot analysis showed a dramatic increase in inflammasome NLRP3 (Figure 3A,C) and microglia/macrophages marker Iba1 (Figure 3B,D) in the injured spinal cord. As expected, the expression levels of both molecules were significantly lower in CD11b KO mice compared to their WT counterparts, indicating that CD11b plays a key role in the inflammatory response and microglia/macrophages activation process. Together, these results demonstrate that CD11b is critical to promoting microglia proliferation, infiltration of peripheral immune cells, and ROS production after SCI.

### 3.3. CD11b KO Mice Show Robust Changes in Neuroinflammation-Related Genes after SCI

To determine the consequences of CD11b deletion on acute inflammatory response following SCI, we evaluated spinal cord tissue at the injury site with NanoString Neuroinflammation panel. Partial least square discrimination analysis (PLS-DA) was used on all normalized transcription count data to reveal a distinct separation between samples of each group, which was clustered into four quadrants (Figure 4A). The two main variants of the PLS-DA model separated samples by injury (variate 1) and genotype (variate 2), accounting for 52% and 10% of the total variation between samples, respectively. Furthermore, the gene *Itgam* regulating expression of CD11b showed significant reduction in the spinal cord of KO mice compared to their WT littermates, which further confirms the validity of the global KO model (Figure 4B). Pairwise comparison between groups yielded a total of 95 (20 downregulated, 75 upregulated) in sham/CD11b KO vs. sham/WT and 123 (54 downregulated, 69 upregulated) genes in SCI/CD11b KO vs. SCI/WT, which indicated robust genotype effects in both baseline and after SCI conditions (Figure 3C,D). Interestingly, more than double the number of genes showed genotype-induced downregulation in SCI mice than sham groups. In terms of injury effects, we were able to observe a total of 390 (175 downregulated, 215 upregulated) when comparing 1d SCI vs. sham in WT (Appendix A), as well as 385 (184 downregulated, 201 upregulated) genes in SCI/CD11b KO vs. sham/CD11b KO groups (Appendix A). As expected, the majority of genes related to neuroinflammation showed dramatic increase at 1d post-injury. Next, we sought to find differentially expressed genes (DEGs) between groups, which yielded the top 20 DEGs in the SCI/CD11b KO vs. SCI/WT comparison set ranked by *p*-value (Figure 4E). Amongst the top DEGs with the lowest *p*-value, *Itgam* and *Mapk14* are involved in the regulation of innate immune response, both of which showed significant reduction in CD11b KO mice after SCI. For cellular function, genes involved with microglia (*Stmn1*, *Dst*) are also downregulated in the injured spinal cord of CD11b KO mice, while the gene *Dlg1* (neurons and neurotransmission) and *Opalin* (oligodendrocyte function) followed similar trends. In addition, four genes (*Apc*, *Agt*, *Fkbp5*, and *Cp*) are involved with the regulation of astrocyte function. Moreover, the genes *Apoe* and *Ep300* are regulators of lipid metabolism, which show significant upregulation in the spinal cord of CD11b KO mice. There are also several genes that are part of the epigenetic regulation process: *Smarca5*, *Kdm4a*, *Eif1*, *Brd4*, and *Smarca4*. Finally, the genes *Cd47* and *Ms4a4a*, both of which show significant upregulation in the KO mice, are enriched on the surface of neurons and microglia, respectively.

To confirm the changes observed in NanoString, we performed validation on several DEGs with qPCR. We first examined the mRNA expression levels of four genes that had significant downregulation in NanoString analysis. *Opalin*, a marker for oligodendrocytes [64,65], showed significant injury-induced downregulation in WT mice (Appendix A) and even lower levels in CD11b KO mice after SCI. The gene *Fcrls,* which modulates Fc receptor-like protein 2, only showed a significant injury-induced upregulation in WT mice (Appendix A) but not in CD11b KO mice, which was markedly lower at the same timepoint after SCI. The genes *Pllp* and *Ennp6* showed significant downregulation after SCI in both WT and CD11b KO, with no significant differentiation between the two genotypes after injury (Appendix A). We next examined three DEGs that were upregulated in NanoString analysis; the first was *Plekhp1,* which showed significant upregulations after injury but no differences between genotypes (Appendix A). In the case of the genes *H2-T23* and *Fkbp5*, SCI/CD11b KO mice showed significant increases at 1 d SCI compared to sham groups (Appendix A), which was also significantly higher than the SCI/WT group.

NanoString pathway enrichment analysis of DEGs showed interferon alpha response as the top pathway upregulated in the SCI/CD11b KO vs. SCI/WT comparison set, with interferon gamma response, inflammatory response, apoptosis, and TNF-alpha signaling via NF-kB as part of the top enriched pathways (Figure 5A). Within the top enriched pathway of interferon alpha response, genes included *Ifitm2*, *Rsad2*, *Irf1*, *Ifih1*, *Cd47*, and *Gbp2* (Figure 5B). Pathway enrichment analysis of downregulated DEGs in SCI/CD11b KO vs. SCI/WT comparison yielded E2F targets as the top pathway (Figure 5C). Other downregulated pathways include UV response Dn, hypoxia, reactive oxygen species pathway, and apoptosis. Specific DEGs within the top downregulated pathway include *Xrcc6*, *Prkdc*, *Stmn1*, *Rpa1*, and *Rad1* (Figure 5D). Due to the critical role that CD11b plays in the acute immune response to SCI, it is reasonable to assume that the pathways activated by injury in WT mice may be different than those activated CD11b KO mice. Based on this hypothesis, we next examined the injury-induced DEGs in WT and CD11b KO groups through pathway enrichment analysis. For WT mice, SCI led to the upregulation of genes in the pathways of interferon gamma response, inflammatory response, IL-6/JAK/STAT3 signaling, TNF-alpha signaling via NF-kB, and apoptosis (Appendix A). Moreover, SCI also led to downregulation of reactive oxygen species pathway, E2F targets, apoptosis, TGF-beta signaling, and UV response Dn in WT mice (Appendix A). On the other hand, in CD11b KO mice, the top upregulated pathway is IL-6/JAK/STAT3 signaling, while the top downregulated pathway is UV response Dn (Appendix A). For analysis of classic pro- and anti-inflammatory genes, we examined eight pro-inflammatory genes (*CD68*, *CD86*, *Cxcl10*, *Il1b*, *Il6ra*, *Tnfrsf10b*, *Tnfrsf1a*, *Tnfrsf1b*) and seven anti-inflammatory genes (*Ccl5*, *Cx3cr1*, *Il10rb*, *Il2rg*, *Tgfb1*, *Tgm2*, *Vegfa*). These genes showed significant upregulation after injury, but no genotype differences were observed (Appendix A–C).

To further investigate the effects of CD11b ablation on the transcriptomic profile of injured spinal cords, we used bulk RNA sequencing to examine spinal cord tissue at 1d post-injury. Using a cutoff point of FDR < 0.05, we were able to screen a total of 137 DEGs (59 downregulated and 78 upregulated) from SCI/CD11b KO vs. SCI/WT and depict them in a volcano plot (Figure 6A). In addition, our analysis also yielded the top 10 DEGs (Figure 6B), with *CD11b*, *Cfh*, *Gpr182*, *Ube2a*, and *Cyp51* being the top 5 genes with the lowest FDR value (Figure 6C). Moreover, pathway enrichment analysis with the Bioplanet

2019 database shows “Interleukin-1 regulation of extracellular matrix” as the top upregulated pathway (Figure 6D), while “Cholesterol biosynthesis” is the top downregulated one in the KO mice (Figure 6E).

In summary, our NanoString and bulk RNA-seq results show upregulation of proinflammatory genes involved with interferon alpha, interferon gamma, and other pathways that participate in the development of innate and adaptive immune responses, suggesting that CD11b KO mice have upregulated acute neuroinflammation in response to spinal cord injury, which is a necessary mechanism of defense for the CNS protection.

### 3.4. Ablation of CD11b Improves Functional Recovery and Reduces Tissue Damage after SCI

Finally, we assessed behavior and histopathology to determine the chronic impact of CD11b ablation on recovery. Weekly assessment of BMS scores and sub-scores showed that SCI/CD11b KO mice were recovering at a faster rate than their WT littermates (Figure 7A,B). Starting from as early as 7 days post-injury, the average score for the SCI/WT mice was 1.11 ± 0.423, indicating that most WT mice within the group only showed slight ankle movement. At the same time point, SCI/CD11b KO mice had an average score of 1.875 ± 0.524, which is indicative of extensive ankle movement but no plantar placement. This dramatic genotype difference at 7 d post-injury (n = 9 for WT, n = 8 for CD11b KO, *p* = 0.025) was the start of a persistent trend that would continue for 6 weeks (*p* < 0.001), after which both injury groups appeared to have reached a plateau for motor function recovery. The average BMS scores at the plateau period was 4.6 for WT mice, indicating that mice had occasional or frequent plantar stepping. In contrast, CD11b KO mice had an average BMS of 5.7, which suggests that animals had frequent or consistent plantar stepping. Moreover, BMS sub-scores also reflected consistent plantar stepping and better coordination. We detected a significant main effect of genotypes [F(1, 15) = 5.549, *p* = 0.033 for BMS scores and F(1, 15) = 5.525, *p* = 0.033 for BMS sub-scores].

Next, we tested whether CD11b is involved in the development of post-injury allodynia evoked by mechanical and thermal stimuli. At 6 w following SCI, mice that regained adequate locomotor function to be able to withdraw a hind paw from a stimulus were selected for further nocifensive behavioral testing. In both tests of nociceptive behavior, there was no difference in mechanical/thermal threshold between the sham groups of either genotype (Figure 7C,D). After SCI, the 50% mechanical pain threshold of SCI/WT was considerably lower than their uninjured counterparts (Figure 7C), whereas CD11b KO mice showed little difference between sham and injury groups. Further examination of results demonstrated that the 50% pain threshold of SCI/CD11b KO mice was significantly higher than SCI/WT mice at 6 w post-injury (Figure 7C), which indicates that hyperesthesia was effectively alleviated by CD11b genetic ablation. In the hot plate test, the temperature threshold for WT mice was significantly decreased after SCI (*p* < 0.05 vs. SCI/WT, Figure 7D), but no differences were observed between the sham and SCI groups of CD11b KO mice, which further confirms the attenuation of SCI-induced allodynia. After completion of the behavioral tests, we examined tissue damage by lesion volume (LV) and spared white matter (SWM). Quantification of lesion volume by unbiased stereology showed a much smaller area of glial scarring in CD11b KO compared to their WT littermates (*p* < 0.01 vs. SCI/WT, Figure 7E,F). Similarly, histological staining with solvent blue showed less myelin damage in SCI/CD11b KO mice compared to WT mice, while quantification showed a significantly higher percentage of SWM (*p* < 0.01 vs. SCI/WT, Figure 7G,H). Further investigation of the injured spinal cord with IHC showed a reduced density of microglia/macrophage Iba1+ and CD86+ cells in the dorsal and lateral WM (Figure 8A–D). Although no genotype differences were observed in the Iba1+ cell numbers in the ventral WM (Figure 8B), the number of CD86+ cells were significantly lower in CD11b KO mice. Examination of the spinal scar with the astrocyte marker GFAP and fibrotic scar marker PDGFrB showed a lower ratio of positive cell areas in the peri-injury region and injury core (Figure 8E–G). Taken together, the results indicate that CD11b depletion improves recovery after SCI, which is associated with reduced tissue damage.

For further assessment of motor coordination, we used Catwalk XT gait analysis to examine fine motor differences beyond that recognizable by BMS scores. Stride length is the distance between successive placements of the same paw (Figure 9A), which showed a significant main injury effect between groups [F (1, 28) = 6.130, *p* = 0.0196], but no main genotype effect. Measurement of print length and width also found significant main injury effects [F (1, 28) = 18.03, *p* = 0.0002 for print length, F (1, 28) = 17.29, *p* = 0.0003 for print width], along with marked decrease in both parameters in WT mice following SCI (*p* < 0.01, Figure 9B,C), but deficits were not significant in CD11b KO mice. Representative footprint images are indicated in Appendix A. Next, we evaluated motor coordination with regularity index, a parameter that tracks the order of paw placement in a step cycle (Figure 9D). A single step cycle is defined as each of the four paws being placed on the walking surface in sequence, which was analyzed by attributing each set of steps into either a normal stepping pattern or abnormal gait. The result is a percentage of normal stepping out of all step cycles analyzed. As expected, the step sequence regularity index was significantly lower in SCI/WT mice compared to sham/WT (*p* < 0.01 vs. sham/WT, Figure 9D and Appendix A), indicating clear deficits in motor coordination, but this decrease was not significant in CD11b KO mice. Phase dispersions, a parameter that describes the temporal relationship between placement of two paws within a step cycle, was used to measure inter-paw coordination (Figure 7E,F). Assessment of the phase dispersions between right forepaw (RF) and left hind paw (LH) yielded significant increase in WT mice following SCI (*p* < 0.001, Figure 9E), but the deficits in CD11b KO were not significant. On the other side, which examines the diagonal dispersion of left forepaws (LF) and right hind paws (RH), deficits from SCI could be observed in both WT (*p* < 0.001) and CD11b KO mice (*p* < 0.05, Figure 9F). Moreover, both parameters showed significant main injury effects [F (1, 28) = 18.58, *p* = 0.0002 for RF- > LH; F (1, 28) = 31.56, *p* < 0.0001 for LF- > RH]. Print position is defined as the distance between a pair of hind paw and forepaw of the same side. Ideally, healthy C57BL/6 mice should be able to place their hind paw next to the location of the forepaw that has just been lifted from the walkway. Following SCI, the print positions of both WT and CD11b KO mice were significantly increased (*p* < 0.001, Figure 9G) while also showing an injury main effect [F (1, 28) = 44.44, *p* < 0.0001]. However, pairwise comparison of SCI/WT and SCI/CD11b KO yielded no statistical significance. Hindlimb base-of-support is a parameter that measures the average width of the track (distance between RH and LH) made by the animal, in which the farther apart the feet are placed during locomotion, the less likely the animal is to fall and the larger the base-of-support (BOS). Following SCI, WT mice showed significant decrease in hindlimb BOS (*p* < 0.05, Figure 9H), indicating a lack of coordination and trunk stability. We next examined print area, max contact area, and max contact max intensity, which could reflect spontaneous pain activity in mice (Figure 9I,J). The SCI/WT group showed a significant decrease in all three parameters, which suggests reduced contact with the Catwalk surface and the potential presence of spontaneous pain. However, no significant decrease was observed in CD11b KO mice, which suggests a lack of spontaneous pain and coincides with our results in mechanical and thermal allodynia.

In order to obtain an overview of the effects that SCI and genotype may have on Catwalk gait analysis, we used PLS-DA to examine their respective contributions to clustering of the gait parameters. The cluster diagram shows that injury contributed 69% to the differential variation, whereas genotype contributed 14% (Appendix A). Furthermore, pairwise Spearman correlation analysis on all Catwalk parameters showed high levels of correlation with injury severity and BMS scores (Appendix A). Based on the correlation coefficients, the parameters of print length (PL), print width (PW), step sequence regularity index (SSRI), and base of support (BOS) have high levels of (coefficient higher than 0.7) positive correlation with endpoint BMS scores, indicating that mice with higher BMS scores would also perform better in these gait parameters. On the other hand, both sets of phase dispersions (PD) and print position (PP), along the categorial scores of injury (0 = sham, 1 = SCI) have a negative correlation with BMS, indicating that injury severity would impact these scores negatively. In addition, the categorical score of genotype (0 = WT, 1 = CD11b KO) shows a negative correlation with Catwalk parameters, suggesting that CD11b KO mice have better recovery of gait. These data suggest CD11b KO mice could recover locomotor function more fully than WT mice.

## 4. Discussion

In the present study, we examined the role that the major integrin CD11b/CD18 (Mac-1, CR3, αMβ2) plays in the acute inflammatory response after SCI, along with its contribution to locomotor dysfunction and neuropathic pain. Our findings show that experimental SCI causes pro-inflammatory activation of microglia/macrophage, characterized by qPCR, flow cytometry, NanoString, and bulk RNA-seq analysis. Genetic deletion of CD11b gene further increases the neuroinflammation in the injured spinal cord tissue. Importantly, CD11b KO mice show significantly improved locomotor functional recovery and reduced cutaneous hypersensitivity to mechanical and thermal stimuli, which were associated with reduced tissue damage. Thus, our data suggest that the CD11b gene may play a deleterious role in the recovery of neurological functions following SCI by suppressing the beneficial acute inflammation.

Using qPCR, we first observed a significant increase in CD11b mRNA in the acute and sub-acute phase of SCI. This is consistent with studies that examined the temporal activation of microglia and macrophages after SCI [18,66]. Studies showed that on d1 after SCI, post-traumatic neuroinflammation mainly involves neutrophils, while microglia/macrophages start to play a more prominent role between d3 and 7 following injuries [67,68]. The ubiquitous nature of CD11b in all types of myeloid cells required examination of both stages following SCI. Thus, we utilized qPCR and other methods to assess the transcriptomic profile of the injured spinal cord at 1d post-injury. Furthermore, we used flow cytometry due to the heterogeneity of immune cells activated at d3 after SCI.

Using CD11b KO mice, we observed higher expression levels of genes involved in the activation of myeloid cells and their associated inflammatory responses. One of the genes that we examined, *Cxcl10*, regulates a chemokine that actively participates in the initiation of microglial activation following injury [60]. Studies have shown that microglial activation in the early stages of SCI has beneficial effects in debris clearance, working with reactive astrocytes to form a glial scar at the lesion site and promoting neuronal regrowth [69,70,71]. We further observed lower expression levels of *Trem2* and *Gdf15* mRNA in the lesion area of CD11b KO mice. High levels of *Trem2* have been reported to be a sign of poor prognosis in Alzheimer’s disease (AD) and a mediator of microglia hyperactivation in neuropathic pain [72,73,74], whereas *Gdf15*, a mediator of macrophage inhibitory cytokine-1 (MIC-1), has been reported to an important marker of mortality in the aged population and stroke patients [75,76,77]. Although reports have also identified myeloid *Gdf15* as a potential mediator for regenerative inflammation [78,79], the acute timepoint of our data excludes regeneration from the scope of study. Taken together, these results suggest that microglia/macrophage in CD11b-deficient mice demonstrate an inflammation-resolving phenotype that facilitates debris clearance.

The surface adhesion receptor CD11b is routinely used in flow cytometry and other techniques as a well-known marker of the microglia/macrophage population. In the current study, the genetic ablation of CD11b/CD18 has created a unique challenge for characterizing the neuroimmune profile of the injured spinal cord. Using a combination of microglia and monocyte markers, we were able to devise a novel gating strategy for identifying the cell populations. In this gating strategy, antibodies against CX3CR1 were used to differentiate resident microglia from infiltrating monocytes. Furthermore, differentiation of monocytes and neutrophils was achieved using Ly6C, a surface marker expressed in most blood-borne cells of monocytic origin [80]. Using Ly6C and Ly6G, Saiwai et al. were able to identify a population of infiltrating monocytes that promotion inflammatory resolution after SCI [81]. Similarly, we used Ly6C and Ly6G to separate monocytes from neutrophils. Following initial gating of Ly6C, we used Cx3CR1 to isolate resident microglia from macrophages. Early studies that utilized in situ hybridization demonstrated that the majority of Cx3CR1+ cells in the spinal cord of experimental autoimmune encephalomyelitis (EAE) rats were microglia [63]. The high specificity of Cx3CR1 to microglia is further confirmed by later studies that developed Cx3CR1-cre lines as a tool for genetic manipulation of microglia [82,83].

By using flow cytometry to examine the injured spinal cords’ inflammatory profile, our data also demonstrate that CD11b KO mice have less microglia accumulation and reduced infiltration of macrophages and neutrophils from the blood circulation following SCI, along with lower ROS production. To this end, we first show that SCI/CD11b KO have significantly lower expression levels of *Trem2* than their WT counterparts. The *Trem2* gene modulates the expression of triggering receptor expressed on myeloid cells-2 (Trem2), which has been shown to be upregulated in the serum of SCI patients [84], and controls microglia hyperactivation in a model of progranulin deficiency [74], along with the transition of microglia towards a proinflammatory phenotype and exacerbation of neuropathic pain [72]. Moreover, a study comparing the mechanisms of inflammatory responses between traumatic and neurodegeneration injuries found that Trem2 plays a major role in the determination of phenotype [85]. In our qPCR results, we also observed downregulation of the oligodendrocyte marker *Opalin*, which is required for promotion of oligodendrocyte growth and differentiation [64,65]. One possible explanation for decreased *Opalin* levels is enhanced phagocytosis and cleanup of damaged cells at the acute stage of SCI.

Consistent with the results from flow cytometry, both NanoString and RNA-seq showed that CD11b KO mice exhibited significant upregulation of pro-inflammatory transcriptomic factors and genes involved in phagocytosis. Pathway enrichment analysis reveals interferon alpha as the top pathway involved in upregulated DEGs for SCI/WT vs. SCI/CD11b KO. A recent study on chronic constriction injury in rats demonstrated that interferon alpha can improve their mechanical pain threshold, possibly having an analgesic effect [86]. Amongst the genes upregulated in SCI/*CD11b KO* mice compared to their WT littermates, Cd47 has been to be instrumental to the phagocytosis engulfment of apoptotic neurons, as CD47 KO mice exhibit excessive synaptic pruning [87]. Moreover, the downregulation of E2F targets coincides with our lab’s prior findings on cell cycle-related genes and neuroinflammation after SCI [88]. Combined with the attenuation of allodynia observed in the present study, these results provide a possible mechanism for neuroprotection in CD11b KO mice. Thus far, we are the first to study the consequences of CD11b genetic ablation in a rodent model of traumatic SCI. In other models of neuronal injury, studies have shown that CD11b is critical in the development of experimental autoimmune encephalomyelitis and myelin phagocytosis [89,90]. In neutrophils, studies show that CD11b regulates adhesion and recruitment in pathologic inflammation [91,92].

Unfortunately, the ubiquitous expression of CD11b in CNS immunity cells has made it difficult for our study to focus on one specific cell type, highlighting an innate limitation. In general, both microglia and macrophages play critical roles in inflammatory responses following trauma. However, it is hard to distinguish the role that CD11b plays in different cell types with a global knockout model. Such differentiation can only be performed with CD11b-flox mice, which we plan to incorporate in future studies focusing primarily on cell-type-specific mechanisms.

Despite advancements in emergency medicine and higher survival rates, no effective therapy has emerged for SCI patients, who often suffer from temporary or permanent motor and sensory deficits. Immunotherapy strategies have achieved exciting progress in treating many CNS degenerative disorders such as Alzheimer’s disease (AD) [93,94,95], Parkinson’s disease (PD) [96,97,98], and multiple sclerosis (MS) [99,100,101], and have even been adapted for traumatic brain injury [102]. Our study shows for the first time that genetic ablation of CD11b could suppress oxygen free radical-associated tissue injury by microglia/macrophages/neutrophils while promoting neuroprotection. Although blockade of CD11b activity through use of antibodies has been reported in other disease models, with one recent study by Wolf et al. showing that ligand-specific blockade of CD11b with CD40L could target regional inflammation [103], such therapeutic methods has yet to be employed in the context of traumatic SCI. Thus, one possible future direction will be the intrathecal injection of ligand-specific antagonists, which has more therapeutic and translational value. Based on the positive results shown in this study, delayed administration of CD11b antagonists may attenuate allodynia following SCI. Ultimately, it will be intriguing to see whether pharmacological manipulation of CD11b/Mac-1 has the same beneficial effects as genetic ablation.

## 5. Conclusions

Taken together, we showed that CD11b KO significantly altered the ability of microglia/monocytes to migrate and infiltrate into the lesion site following SCI. We further demonstrated that SCI/CD11b KO mice had a pro-inflammatory profile and enhanced phagocytic activity with lower ROS production, thus promoting functional recovery following SCI. This was further supported by results from the NanoString Neuroinflammation panel and bulk RNA-seq. In chronic SCI, behavior assessment showed reduced motor deficits in CD11b KO mice compared to their WT littermates, along with mechanical and thermal allodynia attenuation. These findings suggest that CD11b is a major pathophysiological factor in SCI-mediated inflammatory response and related neurological impairments.

## Figures and Tables

**Figure 1 cells-13-01584-f001:**
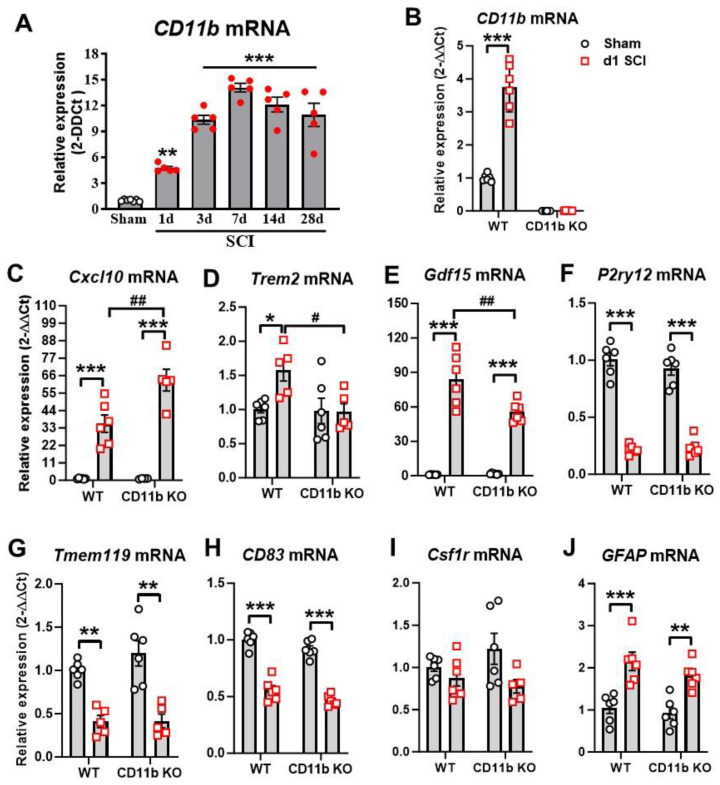
CD11b mediates acute inflammatory response to SCI. (**A**) qPCR analysis was performed to examine the mRNA expression of CD11b in the injured spinal cord of C57BL/6 WT female mice. n = 7 (sham) and 5 (SCI). ** *p* < 0.01, *** *p* < 0.001 vs. sham group. One-way ANOVA following Dunnett’s multiple comparisons test. (**B**–**J**) qPCR analysis was used to examine inflammatory responses in the injured spinal cord of young adult CD11b KO and WT female mice at 1d SCI. * *p* < 0.05, ** *p* < 0.01, *** *p* < 0.001 vs. sham/WT. # *p* < 0.05, ## *p* < 0.01 vs. SCI/WT. n = 5–6/group. Two-way ANOVA followed by Tukey’s post hoc test.

**Figure 2 cells-13-01584-f002:**
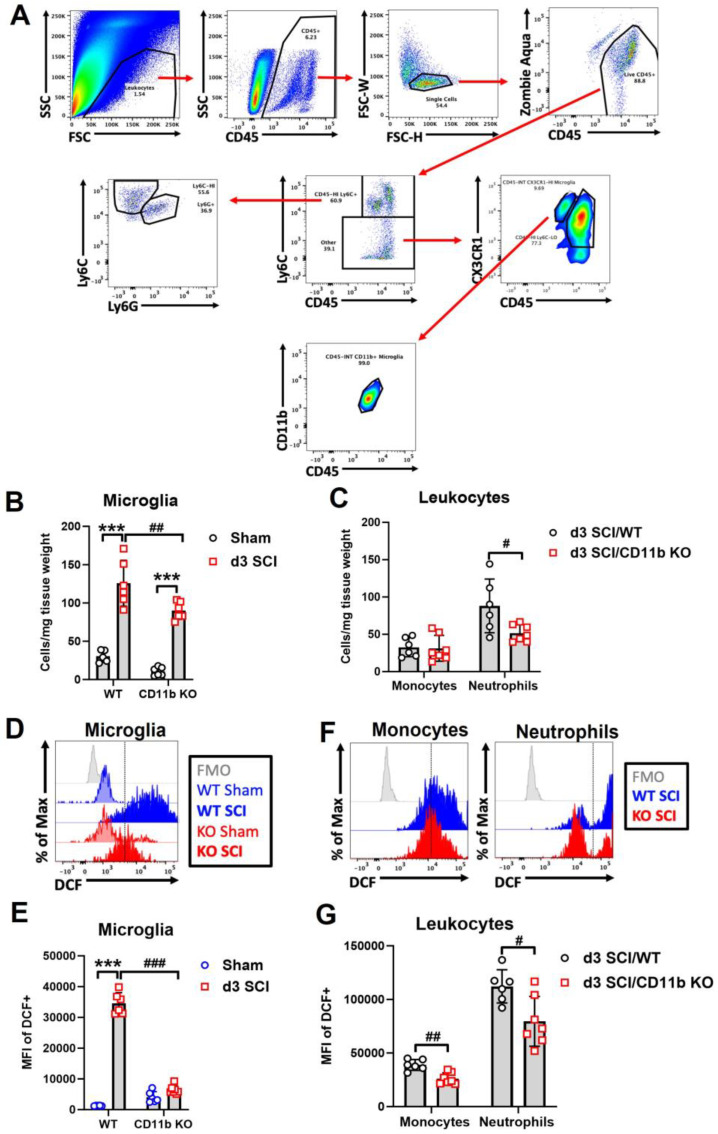
Genetic ablation of CD11b attenuates pro-inflammatory activation of microglia and monocytes at 3 days post-injury. (**A**) Gating strategy for microglia (CD45^int^Ly6C^−^CX3CR1^+^), monocytes (CD45^hi^Ly6C^+^Ly6G^−^), and neutrophils (CD45^hi^Ly6C^+^Ly6G^+^) in CD11b KO mice. (**B**,**C**) Quantification of microglia (**B**) and myeloid (**C**) cell counts are shown. (**D**–**G**) Reactive oxygen species (ROS) and oxidative stress were measured using DCF. Main effects of genotype and injury are seen in the ROS production levels of microglia (**D**,**E**) and myeloid cells (**F**,**G**). *** *p* < 0.001 vs. sham groups. # *p* < 0.05, ## *p* < 0.01, ### *p* < 0.001 vs. SCI/WT. n = 5–6 mice/group. Two-way ANOVA followed by Tukey’s post hoc test.

**Figure 3 cells-13-01584-f003:**
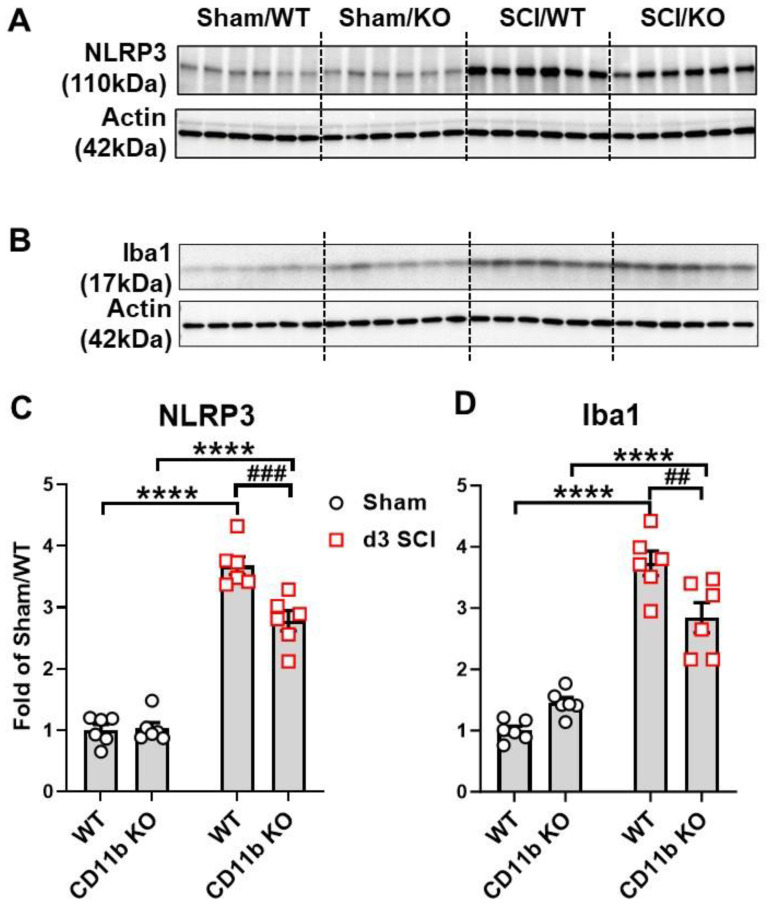
Genetic ablation of CD11b suppresses NLRP3 expression and Iba1 microglia/macrophages activation at 3 days post-injury. (**A**,**B**) Western blotting (WB) images show expression of inflammasome marker NLRP3 and microglia/macrophages marker Iba1. (**C**,**D**) WB quantification. **** *p* < 0.0001 vs. sham groups. ## *p* < 0.01, ### *p* < 0.001 vs. SCI/WT. n = 6/group. Two-way ANOVA followed by Tukey’s post hoc test.

**Figure 4 cells-13-01584-f004:**
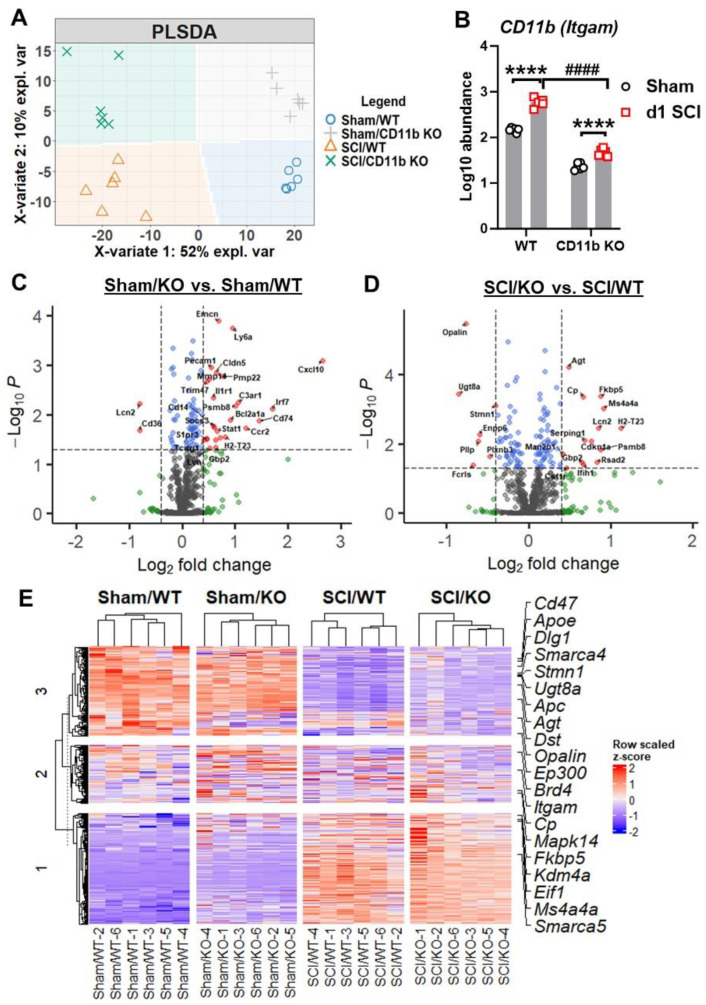
NanoString analysis of spinal cord tissue is performed at 1 day after SCI. (**A**) Partial least square differentiation analysis (PLS-DA) was performed with all normalized gene counts from the NanoString neuroinflammation panel. The two main components of variation were captured on the x- and *y*-axis and show a clear separation of clusters between the injury and genotype groups. (**B**) Log10 abundance of the *Itgam* (CD11b) gene was obtained from NanoString transcriptional data. **** *p* < 0.0001 vs. sham groups. #### *p* < 0.0001 vs. SCI/WT. n = 6/group. Two-way ANOVA followed by Tukey’s post hoc test. (**C**,**D**) Volcano plot of differentially expressed genes (DEGs) in the spinal cord of sham (**C**) and SCI (**D**) spinal cord samples with log2(fold-change) larger than 0.4 and log10(P) higher than 1.3 at the cutoff points depicted by dashed lines. (**E**) Heatmap of the top 20 DEGs in pairwise comparison between SCI/*CD11b KO* and SCI/WT. n = 6/group.

**Figure 5 cells-13-01584-f005:**
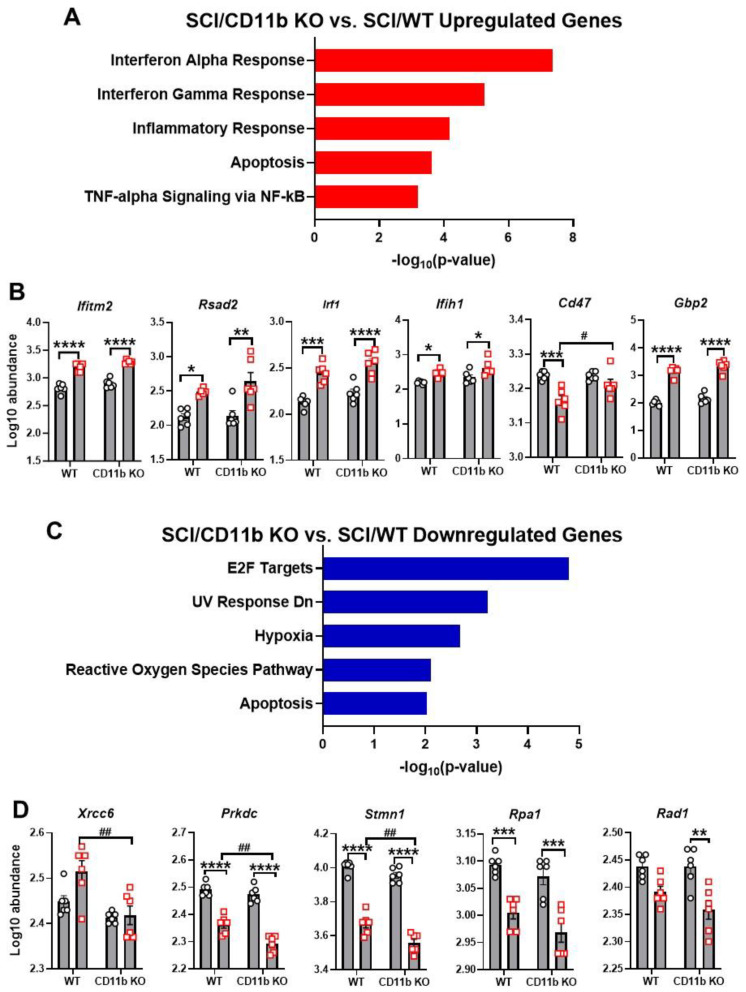
Enrichment analysis reveals potential pathways involved with CD11b-mediated neuroinflammation. (**A**,**B**) Pathway enrichment analysis of upregulated DEGs (**A**) with MSigDB Hallmark 2020 and genes involved with interferon alpha response (**B**). (**C**,**D**) Pathway enrichment analysis of downregulated DEGs (**A**) and genes involved with E2F targets (**D**). n = 6 mice/group. * *p* < 0.05, ** *p* < 0.01, *** *p* < 0.001, **** *p* < 0.0001 vs. sham groups. # *p* < 0.05, ## *p* < 0.01, vs. SCI/WT. n = 6 mice/group, two-way ANOVA followed by Tukey’s post hoc test.

**Figure 6 cells-13-01584-f006:**
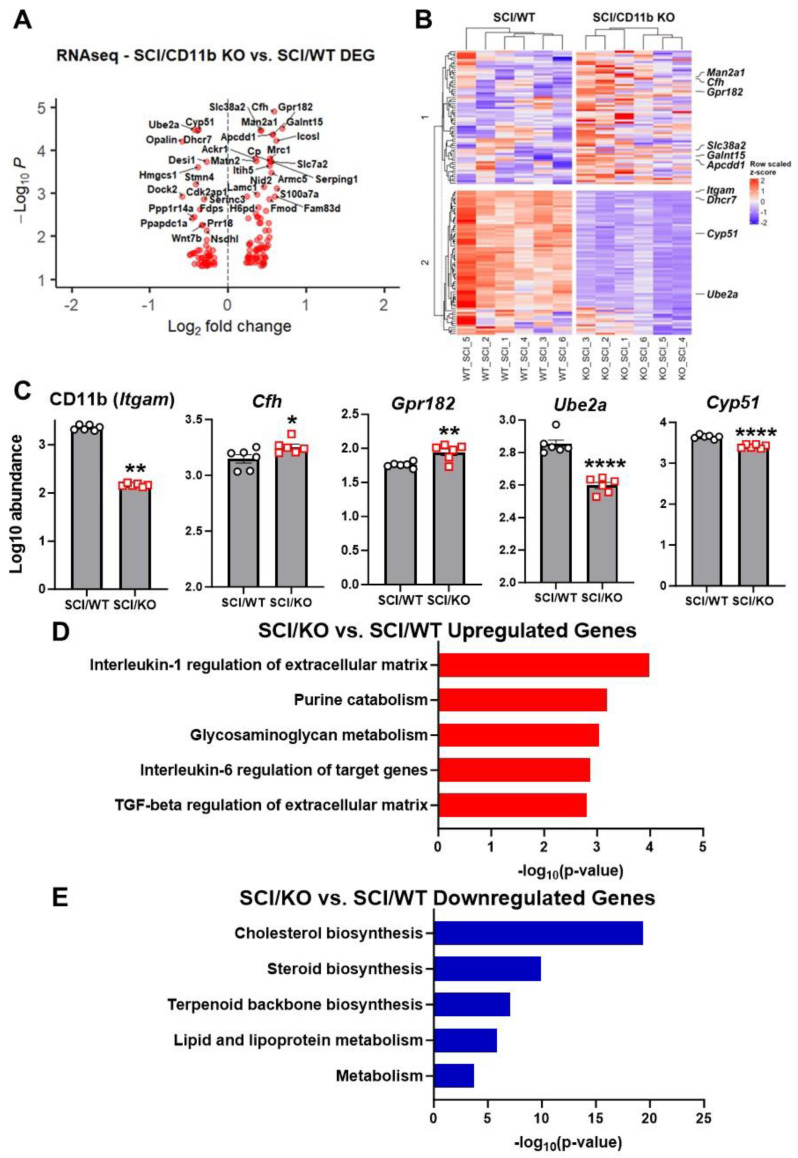
RNA sequencing of injured spinal cord samples at 1d SCI. (**A**) Volcano plot of DEGs obtained from bulk RNA-seq. (**B**) Heatmap of top 10 DEGs in SCI/CD11b KO vs. SCI/WT. (**C**) Log10 abundance of top 5 DEGs obtained from pairwise comparison. (**D**,**E**) Pathway enrichment analysis of up- and downregulated genes. * *p* < 0.05, ** *p* < 0.01, **** *p* < 0.0001 vs. SCI/WT. n = 6 mice/group, two-way ANOVA followed by Tukey’s post hoc test.

**Figure 7 cells-13-01584-f007:**
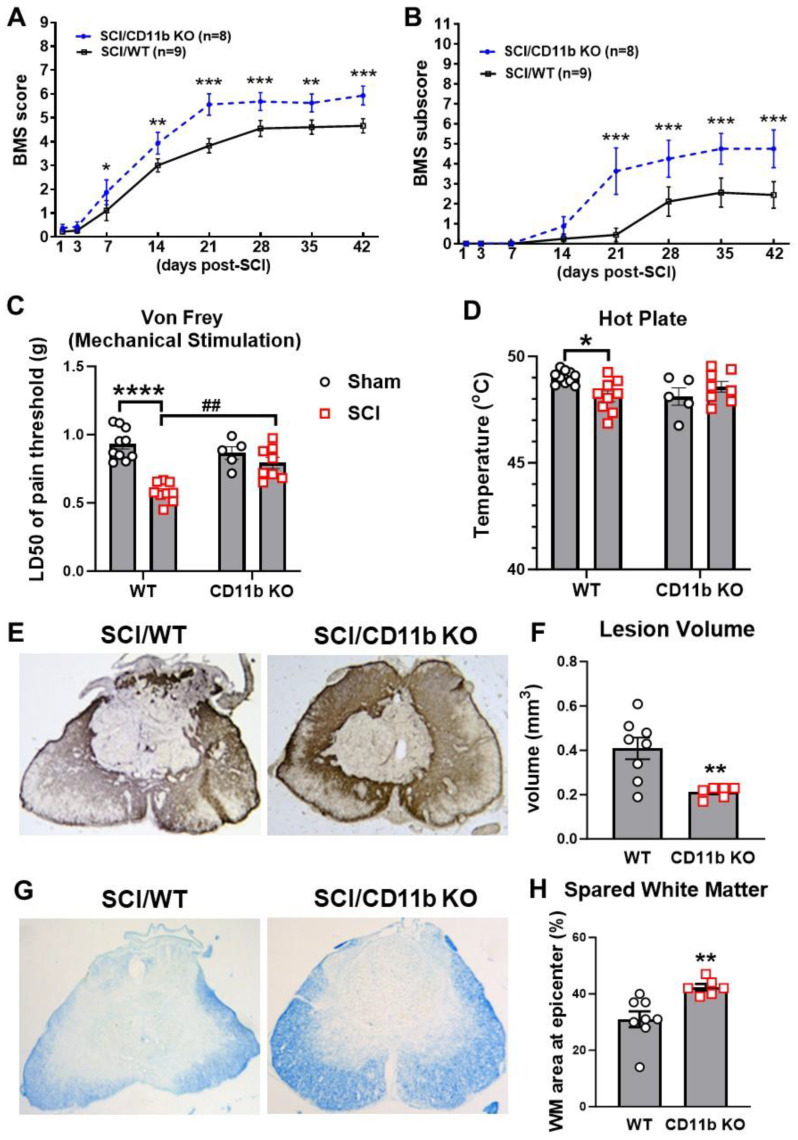
CD11b ablation improves motor and sensory functional recovery following SCI. (**A**,**B**) BMS scores and sub-scores were recorded weekly to quantify hindlimb locomotor recovery after SCI. * *p* < 0.05, ** *p* < 0.01, *** *p* < 0.001 vs. SCI/WT. n = 8–9 mice/group. Repeated two-way ANOVA followed by Holm–Sidak post hoc analysis. (**C**,**D**) von Frey monofilament stimulation and hot plate tests were utilized at 7 weeks post-injury for assessment of mechanical and thermal allodynia. * *p* < 0.05, **** *p* < 0.0001 vs. sham/WT. ## *p* < 0.01 vs. SCI/WT. n = 5 for sham/CD11b KO and 8–10 for other groups. Two-way ANOVA followed by Tukey’s post hoc test. (**E**,**F**) Representative images and quantification of lesion volume (LV) at 8 w SCI via GFAP-DAB staining. (**G**,**H**) Representative images and quantification of spared white matter (SWM) at 8 w SCI. ** *p* < 0.01 vs. SCI/WT. n = 6–8 mice/group. Unpaired *t*-test.

**Figure 8 cells-13-01584-f008:**
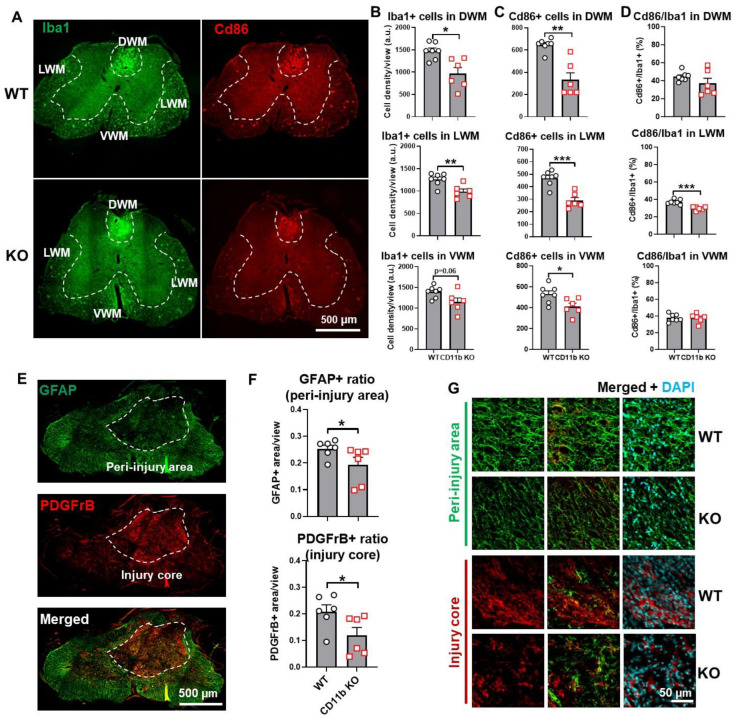
Chronic neuroinflammation and formation of fibrotic scars was alleviated in CD11b-KO mice after SCI. (**A**) IHC representative images of Iba1+(green)/Cd86+(red) cells at 0.4–0.6 mm rostral to the epicenter. Areas of dorsal white matter (DWM), lateral white matter (LWM), ventral white matter (VWM) are outlined for quantification. (**B**) KO mice exhibit significantly reduced Iba1 positive cells in DWM and LWM, not VWM after SCI. (**C**) Significantly decreased Cd86 positive cells are detected in all three regions in KO SCI mice. (**D**) The ratio of Cd86 and Iba1 double-positive cells show significant difference only in LWM between the two SCI groups. (**E**) Representative images demonstrate that PDGFrB (red) is mainly distributed inside the injury core while GFAP in the peri-injury area. (**F**) Either the relative GFAP-positive area or the relative PDGFrB-positive area are significantly lowered in KO mice in the two regions, respectively. (**G**) Representative images display the visual difference from GFAP or PDGFrB imaging between the two SCI groups in the peri-injury area or injury core. * *p* < 0.05, ** *p* < 0.01, *** *p* < 0.001 vs. SCI/WT. n = 6–7/group. Unpaired *t*-test. Scale bars = 500 µm (**A**,**E**) and 50 µm (**G**).

**Figure 9 cells-13-01584-f009:**
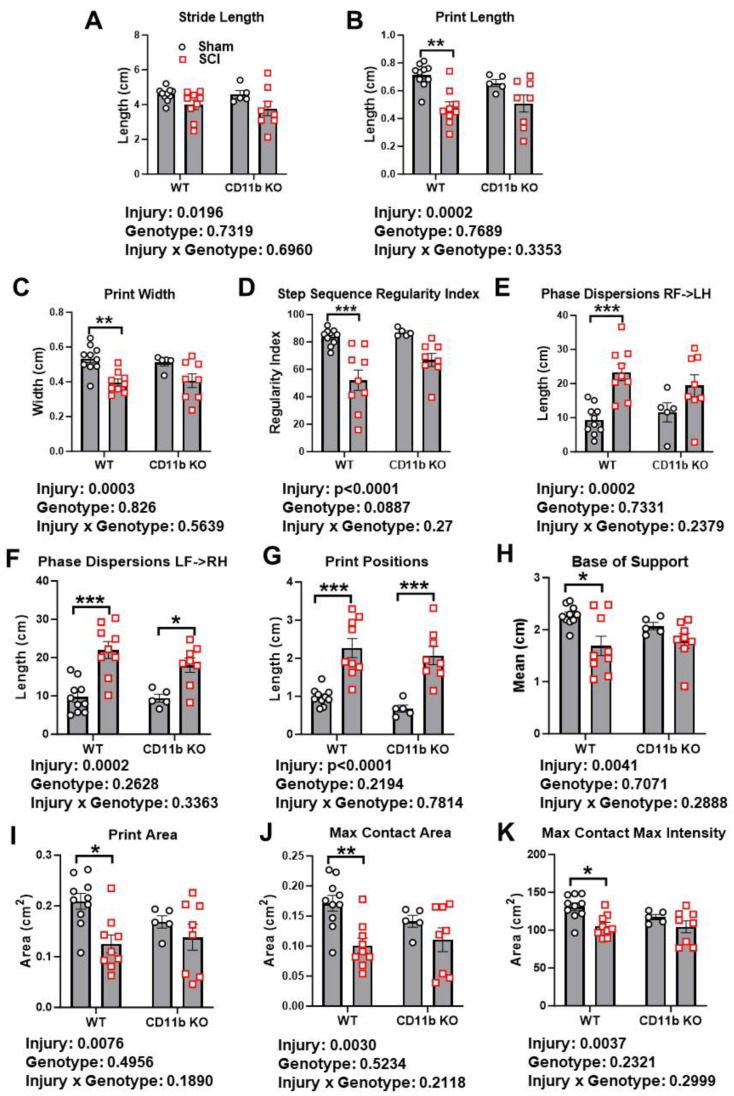
CD11b KO mice showed less deficits of motor coordination after chronic SCI. At 7 weeks post-injury, Catwalk automated gait analysis was used to assess stride length (**A**), print length (**B**), print width (**C**), step sequence regularity (**D**), phase dispersions (**E**,**F**), print positions (**G**), base of support (**H**), print area (**I**), max contact area (**J**), and max contact max intensity (**K**). * *p* < 0.05, ** *p* < 0.01, *** *p* < 0.001 vs. sham groups. n = 5 for Sham/CD11b KO, n = 8–10 for other groups. Two-way ANOVA followed by Tukey’s post hoc test.

## Data Availability

All data needed to evaluate the conclusions in the paper are present in the paper and/or the Appendix A.

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
