# Peer review of "Ablation of the Integrin CD11b Mac-1 Limits Deleterious Responses to Traumatic Spinal Cord Injury and Improves Functional Recovery in Mice"

_cells, 2024, doi:10.3390/cells13181584_

Round 1

Reviewer 1 Report

Comments and Suggestions for Authors

This study tested the potential role of integrin CD11b in neuroinflammation and functional recovery after traumatic spinal cord injury (SCI). While CD11b is widely used as a marker for neuroinflammation after neurological disorders, its role in spinal cord injury remains to be defined. This study used CD11b knockout mice to provide valuable insights into the significant roles of CD11b in neuroinflammation and, importantly, functional deficits after SCI. Combined with previous studies using CD11b blocking antibodies, this study supports targeting CD11b as a potential treatment for SCI. The significance of this study is high. The experiments are well designed, using comprehensive approaches to assess neuroinflammation and functional recovery following SCI. The manuscript is overall well-written. However, there are some concerns that need to be addressed to further improve this interesting study.

1. The flow cytometry data at 3 days after injury show that CD11b deletion reduces the number of microglia and neutrophils in the injured spinal cord. These data indicate that neuroinflammation is decreased by CD11b deletion, which somewhat contradicts the Nanostring and bulk RNA data at day 1 showing CD11b KO mice have upregulated acute neuroinflammation. This raises an interesting question about whether the effects of CD11b deletion on neuroinflammation are immune cell-specific and change over injury time. Some important gene changes observed on day 1 should be validated on days 3 and 7, at least by qPCR.

2. The overall anatomical assessments need to be strengthened. At day 7, when the BMS was first significantly improved in CD11b mice compared to control ones, histology and immunohistology staining for neuroinflammation and neuroprotection should be included. These data will be important to understand how CD11b deletion improves locomotion. At the chronic stage (day 42 after SCI), only GFAP immunohistology at the epicenter (Fig. 6E) is shown. The neuroinflammation, glial responses, and spared gray/white matter around the injury should be shown and quantified.

3. Minor concerns: 1) Although the combination of CX3CR1 with Ly6Clow and CD45 intermediate is reasonable for microglia, lines 288-299, “the cell surface marker CX3CR1 is expressed specifically in microglia,” is overstated since it is also expressed in macrophages at least. 2) although their previous studies are appropriate, a brief description how lesion volume is quantified should be included in methods.

Author Response

Reviewer 1

  1. The flow cytometry data at 3 days after injury show that CD11b deletion reduces the number of microglia and neutrophils in the injured spinal cord. These data indicate that neuroinflammation is decreased by CD11b deletion, which somewhat contradicts the Nanostring and bulk RNA data at day 1 showing CD11b KO mice have upregulated acute neuroinflammation. This raises an interesting question about whether the effects of CD11b deletion on neuroinflammation are immune cell-specific and change over injury time. Some important gene changes observed on day 1 should be validated on days 3 and 7, at least by qPCR.

Response: To validate the d1 NanoString results, we performed Western blot to determine the expression levels of key neuroinflammatory markers in the spinal cord on day 3 after SCI. We found that CD11b KO mice had lower levels of the inflammasome marker NLRP3 and microglia/macrophage Iba1. Results have been added to a new figure in Fig. 3.

  1. The overall anatomical assessments need to be strengthened. At day 7, when the BMS was first significantly improved in CD11b mice compared to control ones, histology and immunohistology staining for neuroinflammation and neuroprotection should be included. These data will be important to understand how CD11b deletion improves locomotion. At the chronic stage (day 42 after SCI), only GFAP immunohistology at the epicenter (Fig. 6E) is shown. The neuroinflammation, glial responses, and spared gray/white matter around the injury should be shown and quantified.

Response: Unfortunately, we don’t have samples available at 7d SCI for histology and immunohistology. In future studies, we will perform more experiments focused on neuroinflammation and neuroprotection in the acute and sub-acute time points. To examine neuroinflammation, Iba1 and CD86 staining were performed to assess microglia/macrophage activation. Furthermore, IHC was also performed on tissue samples at 8w SCI with GFAP for astrogliosis and PDGFrB for fibrotic scarring. Both sets of IHC staining have been added to a new figure in Fig. 8.  As for the analysis of spared white matter at the chronic stage (8w SCI), we have added the images and quantification to Fig.7G-H in the revised manuscript.

  1. Minor concerns: 1) Although the combination of CX3CR1 with Ly6Clow and CD45 intermediate is reasonable for microglia, lines 288-299, “the cell surface marker CX3CR1 is expressed specifically in microglia,” is overstated since it is also expressed in macrophages at least. 2) although their previous studies are appropriate, a brief description how lesion volume is quantified should be included in methods.

Response: We have revised the statement to be more accurate for CX3CR1 cell-type expression into following, “The cell surface marker CX3CR1 is expressed in microglia and macrophages” (page 7, line# 336-7). For the second point, we have added a brief description of how lesion volume was calculated in the methods section, including grid spacing and criteria (page 5, line# 213-6).

Reviewer 2 Report

Comments and Suggestions for Authors

The authors examine the role of CD11b integrin in the inflammatory response and in functional and tissue recovery after spinal cord injury. 

To this end, they use a KO mouse model.

The results presented are interesting and highlight the key role of CD11b in setting off inflammation and subsequently in recovery in injured animals. 

However, I would like to comment on the histological analysis of the spinal cord after SCI. 

The authors use qPCR and flow cytometry to study the inflammatory reaction and glial scar(GFAP), but unfortunately these analyses are not included in the histological analysis. 

The authors should analyse spinal scar in both groups of animals using GFAP (glial scar), PDGFrB (fibrotic scar), Iba1 and CD86/CD206 (for the inflammatory component) in histology and add these results to those shown in figure 6.

I also noted two other minor points:

In the interpretation of the results in figure 1 the authors write: "Collectively, acute SCI leads to increased levels of genes that initiate inflammation-resolving processes and drive myeloid cells towards clearanceof damaged tissue and debris, which was not affected by CD11b ablation". 

I do not agree with this, we can see in Figure 1C, D and E that KO CD11b has an effect on the expression of these three genes compared with the control group. 

In addition, Figure 7 shows that many parameters are unchanged between the groups with and without SCI in KO animals, which is not the case for WT animals. To illustrate this, the authors should carry out a correlation analysis with and without lesion in these two conditions in order to demonstrate more precisely the effect of the KO in these tests. 

Author Response

  1. The authors should analyse spinal scar in both groups of animals using GFAP (glial scar), PDGFrB (fibrotic scar), Iba1 and CD86/CD206 (for the inflammatory component) in histology and add these results to those shown in figure 6.

Response: IHC was performed on tissue samples at 8w SCI with GFAP, PDGFrB, Iba1 and CD86. However, we couldn’t get good staining for CD206. This may be due to the low number of alternatively activated microglia/monocytes in the chronic phase as late as 8w post-injury. Images and relevant quantification have been added to Fig. 8.

minor points:

  1. In the interpretation of the results in figure 1 the authors write: "Collectively, acute SCI leads to increased levels of genes that initiate inflammation-resolving processes and drive myeloid cells towards clearance of damaged tissue and debris, which was not affected by CD11b ablation". I do not agree with this, we can see in Figure 1C, D and E that KO CD11b has an effect on the expression of these three genes compared with the control group. 

Response: We have revised the concluding remark for this section to “which was partially affected by CD11b ablation” (page 7, line# 330).

  1. In addition, Figure 7 shows that many parameters are unchanged between the groups with and without SCI in KO animals, which is not the case for WT animals. To illustrate this, the authors should carry out a correlation analysis with and without lesion in these two conditions in order to demonstrate more precisely the effect of the KO in these tests.

Response: Using a compilation of the Catwalk data featured in Fig. 9, we used Partial least square discrimination analysis (PLS-DA) to calculate the overall percentage of variation that mice genotype would contribute to gait while walking, which has been added to Fig. S6A in supplemental materials. Furthermore, we performed pairwise Pearson correlation analysis on all Catwalk parameters with the added rank variable of Injury (0=Sham, 1=SCI), Genotype (0=WT, 1=CD11b KO), and the final BMS scores at 6w post-injury. Specific correlation coefficients are featured in a heatmap of the correlation matrix in Fig. S6B.

Reviewer 3 Report

Comments and Suggestions for Authors

This paper by Yun Li et al demonstrated that the pathological effects of integrin CD11b mac-1 on traumatic spinal cord injury in mice. The general purpose of this study is clear. The study appears to be of interest, whereas I have some concerns with their results and discussion. Therefore I would recommend it for acceptance after the minor points listed below are addressed.

Comment #1 Inflammation

Although some cytokines (pro-inflammatory cytokines) act to make disease worse, others serve (anti-inflammatory cytokines) to reduce inflammation and promote healing. The authors should also consider and discuss the involvement of pro-inflammatory and anti-inflammatory mediators in the pathogenesis of SCI. I think the authors should analyze these two process separately in the results section and show Fig4, 5 in the revised manuscript.

Comment #2 CD11b positive cell

In generally, CD11b is a marker for macrophages and microglia in immune responses. Unfortunately, I didn't understand which you focused on the microglia and/or macrophages in this study. The authors need to describe or discus this point with evidence in the revised manuscript.

I hope these comments will be helpful.

Author Response

  1. Although some cytokines (pro-inflammatory cytokines) act to make disease worse, others serve (anti-inflammatory cytokines) to reduce inflammation and promote healing. The authors should also consider and discuss the involvement of pro-inflammatory and anti-inflammatory mediators in the pathogenesis of SCI. I think the authors should analyze these two process separately in the results section and show Fig4, 5 in the revised manuscript.

Response: The NanoString results in Fig. 4 and Fig. 5 are based on p-values derived from the pairwise comparison of SCI/WT and SCI/CD11b KO groups. Many of the well-known pro- and anti-inflammatory mediators didn’t pass the initial filter of p-values, so we didn’t show genes categorized in this way. For additional information, we grouped the pro- and anti-inflammatory genes in a supplementary figure (Fig.S3) of the revised manuscript.

  1. In generally, CD11b is a marker for macrophages and microglia in immune responses. Unfortunately, I didn't understand which you focused on the microglia and/or macrophages in this study. The authors need to describe or discuss this point with evidence in the revised manuscript.

Response: We agree that both microglia and macrophages participate in the spinal cord's immunological response to traumatic injury. However, it is hard to distinguish the role that CD11b plays in different cell types with a global knockout. This can only be performed with CD11b-flox mice, which we are actively breeding and plan to incorporate into our future studies. We will have a more definitive answer to this question in our next publication of the project. We have added a paragraph regarding this limitation to the discussion on page 13 (line 638-44) of the revised manuscript.

Round 2

Reviewer 1 Report

Comments and Suggestions for Authors

The authors have addressed this reviewer's concerns!